# The Association of Smoking and Hyperuricemia with Renal Arteriolosclerosis in IgA Nephropathy

**DOI:** 10.3390/biomedicines11072053

**Published:** 2023-07-21

**Authors:** Yuki Shinzato, Ryo Zamami, Nanako Oshiro, Takuto Nakamura, Akio Ishida, Yusuke Ohya, Kentaro Kohagura

**Affiliations:** 1Department of Cardiovascular Medicine, Nephrology and Neurology, Graduate School of Medicine, University of the Ryukyus, Okinawa 903-0215, Japannananew@ymail.ne.jp (N.O.); yaican@med.u-ryukyu.ac.jp (A.I.); 2Dialysis Unit, University Hospital of the Ryukyus, Okinawa 903-0125, Japan; 3University of the Ryukyus Hospital, Okinawa 903-0125, Japan

**Keywords:** smoking, hyperuricemia, renal arteriolosclerosis, IgA nephropathy

## Abstract

The combination effects of smoking (SMK) and hyperuricemia (HU) on renal arteriolosclerosis in patients with IgA nephropathy remain unknown. We examined the cross-sectional association between smoking (current or former) and renal arteriolar hyalinosis and wall thickening with or without HU [uric acid (UA) level ≥ 7 and ≥5 mg/dL in men and women] in 87 patients with IgA nephropathy who underwent renal biopsy. Arteriolar hyalinosis and wall thickening were assessed by the semiquantitative grading of arterioles. The SMK/HU subgroup showed the highest indices for hyalinosis and wall thickening, followed by the non-SMK/HU, SMK/non-HU, and non-SMK/non-HU subgroups. Multiple logistic analysis showed that SMK/HU, but not SMK/non-HU, was significantly associated with an increased risk of higher-grade renal arteriolar wall thickening. However, this did not occur with hyalinosis compared to non-SMK/non-HU. The adjusted odds ratio (95% confidence interval, *p* value) for SMK/HU was 12.8 (1.36–119, *p* < 0.05) for wall thickening. An association between SMK and renal arteriolar wall thickening might be prevalent only among patients with HU and in patients with IgA nephropathy. Further prospective studies are needed to determine whether patients with HU and SMK history exhibit rapid eGFR deterioration.

## 1. Introduction

A meta-analysis of observational studies has suggested that smoking (SMK) has adverse effects on chronic kidney disease (CKD) with considerable heterogeneity [1]. Previous studies have demonstrated that SMK is associated with microalbuminuria [2,3], proteinuria [4], progression to renal failure [5,6], and end-stage renal disease [5], whereas others found no significant association between SMK and CKD progression [7,8]. Similar heterogeneity was found in the association of SMK with renal progression in IgA nephropathy [9,10]. Previous epidemiological studies have suggested that the adverse effects of SMK on kidney function might be more prevalent in conjunction with specific comorbidities [11].

Although an association between SMK and CKD progression has not been clearly demonstrated, hemodynamic and nonhemodynamic factors may play important roles in their underlying mechanism [11]. SMK is suggested to be linked with ischemic nephropathy in patients with peripheral atherosclerotic disease [12]. A previous study reported that nephrosclerosis is the main pathological indicator of CKD in smokers [13]. Moreover, other studies have demonstrated that SMK is a potential risk factor for renal arteriosclerosis among patients with CKD [14] and kidney transplant recipients [15]. Renal arteriolar hyalinosis, one of the types of arteriosclerosis, was suggested to cause CKD progression accompanied by glomerular hypertension via the augmentation of susceptibility to hypertensive renal damage [16]. In IgA nephropathy, SMK was also associated with renal arteriolar hyalinosis [17]. Similar to the animal study [18], we reported an association between hyperuricemia (HU) and arteriolosclerosis (hyalinosis and wall thickening) in patients with CKD who underwent renal biopsy, although a sole association between SMK and these arteriolosclerosis indices was not shown [19]. However, the combination effect of SMK and HU on renal arteriolosclerosis remains unknown. SMK and HU are suggested to cause endothelial dysfunction and vascular smooth muscle proliferation [20,21]. Therefore, we found that there were interactive effects of SMK and HU on renal arteriolar hyalinosis and wall thickening. This study aimed to identify an association between the combination of the SMK status and HU and renal arteriolar hyalinosis and wall thickening in patients with IgA nephropathy: the most common primary glomerular disease.

## 2. Materials and Methods

### 2.1. Settings and Participants

A total of 93 consecutive patients who were newly diagnosed with IgA nephropathy at the University of the Ryukyus Hospital between 1 January 2003 and 31 December 2006 were considered for inclusion in this study. The six patients using urate-lowering drugs were excluded from the analysis. Finally, 87 patients were included in this study.

### 2.2. Semiquantitative Assessment of Preglomerular Arteriole Damage

Using previously described methods [19], we assessed preglomerular arteriolar hyalinosis and wall thickening. Regarding the arteriolar hyalinosis, the hyalinotic lesion of arteriole in each specimen was semi-quantitatively assessed as follows: grade 0 (G0), no hyalinosis of the vessel wall; grade 1 (G1), hyalinosis of <25%; grade 2 (G2), hyalinosis of 25–50%; and grade 3 (G3), hyalinosis of >50% of the vessel wall circumference. We calculated the mean grade of the renal arteriolar hyalinosis (arteriolar hyalinosis index) in each patient with these grading systems using the following formula: arteriolar hyalinosis index = (n0 × 0 + n1 × 1 + n2 × 2 + n3 × 3)/N. Here, n1, n2, and n3 indicated the number of arterioles showing wall thickening scores of G1–G3, and N showed the total number of arterioles. Similarly, we semi-quantitatively assessed the wall thickening of the arteriole in each specimen as follows: grade 0 (G0), no thickening; grade 1 (G1), mild thickening; grade 2 (G2), moderate thickening without definite narrowing of the lumen; and grade 3 (G3), severe thickening with definite narrowing of the lumen. With these grading systems, we calculated the mean grade of the renal arteriolar wall thickening (arteriolar wall thickening index) in each patient using the following formula: arteriolar wall thickening index = (n0 × 0 + n1 × 1 + n2 × 2 + n3 × 3)/N. Here, n1, n2, and n3 indicated the number of arterioles exhibiting wall thickening scores of G1–G3, and N represented the total number of arterioles. The representative microphotographs of each grade of arteriolar hyalinosis and wall thickening are available in our previous report [19]. A physician (K.K.) who was blind to the patient information performed all the histological analyses.

### 2.3. Association of Uric Acid (UA) Levels and Renal Arteriolar Hyalinosis According to SMK Status

We identified sex-specific differences in serum uric acid (UA) cut-off values, which were 5 mg/dL for women and 7 mg/dL for men, which indicated an increased risk of renal arteriolar hyalinosis according to our previous study. Therefore, we defined UA levels as ≥5 mg/dL in women and ≥7 mg/dL in men. The smokers were defined as current or former smokers. Body mass index (BMI) was calculated by dividing the weight in kilograms by the height in meters squared. Dyslipidemia was defined as hyper-low-density lipoprotein (LDL) cholesterolemia, hypertriglyceridemia, hypo-high-density lipoprotein (HDL) cholesterolemia, and/or the use of antidyslipidemia. In addition to a medical history of diabetes, diabetes mellitus was determined by fasting and a postprandial glucose level measurement or by a 75 g oral glucose tolerance test. Urinary protein levels were measured in first-spot morning urine samples. The estimated glomerular filtration rate (eGFR) was calculated using a revised equation for Japanese individuals: eGFR (mL/min per 1.73 m^2^) = 194 × serum creatinine^−1.094^ × age^−0.287^ (×0.739, if female) [22].

### 2.4. Statistical Analysis

A one-way analysis of variance (ANOVA) and the chi-square test were used to analyze differences in discrete variables between the groups. A one-way ANOVA was also used to analyze the interaction effect of SMK and HU on arteriolosclerosis indices. Higher-grade arteriolar hyalinosis and wall thickening were defined as hyalinosis or wall thickening with values equal to or greater than the third tertial hyalinosis index (0.41) and the wall thickening index (0.316). Multivariate logistic regression analysis was employed to determine the relationships between SMK/HU subgroups and arteriolar hyalinosis and wall thickening. We adjusted for age, sex, and potential confounding factors such as hypertension. The use of renin-angiotensin system (RAS) inhibitors was also included because they reversed arteriolar hypertrophy independent of their blood pressure-lowering effect [23,24]. The following covariates were included in the adjusted models: age and sex in model 1, model 1 plus BMI >25, history of diabetes mellitus or hypertension or dyslipidemia and eGFR in model 2 and model 2, plus the use of RAS inhibitors in model 3. Data were expressed as medians with the interquartile range. All probability values were two-tailed, and the significance level was set at a *p*-value of <0.05.

## 3. Results

The SMK/HU subgroup was characterized by older age, a higher prevalence of hypertension, hyperlipidemia, a higher BMI, and lower eGFR (Table 1).

### 3.1. Hyalinosis Index and Rate of Max Grade of Hyalinosis among the Subgroups

We analyzed the indices of hyalinosis for each SMK/HU subgroup. The hyalinosis index in the SMK/HU subgroup was the highest among the groups (Figure 1). The rate of the higher max grade for the hyalinosis index (G2 + G3) was the highest in the SMK/HU subgroup (Figure 2).

### 3.2. Wall Thickening Index and Rate of Max Grade of Wall Thickening among the Subgroups

We analyzed the indices of wall thickening for each SMK/HU subgroup. The wall thickening index in the SMK/HU subgroup was the highest among the groups (Figure 3). The rate of the higher max grade for the wall thickening index (G2 + G3) was the highest in the SMK/HU subgroup (Figure 4).

### 3.3. The Risk of Higher-Grade Arteriolar Hyalinosis and Wall Thickness

We analyzed the odds ratios (ORs) of various risk factors for higher-grade arteriolar hyalinosis. Age, the history of hypertension or diabetes mellitus or dyslipidemia, eGFR (high to low), SMK/HU, and non-SMK/HU (non-SMK/non-HU was used as a reference) were significantly associated with a higher grade of arteriolar hyalinosis in the non-adjusted model. The association between non-SMK/HU and arteriolar hyalinosis was significant in multivariate model 3, which included age, sex, potential risk factors such as hypertension, and the use of RAS inhibitors (Table 2).

We also analyzed the ORs of different risk factors for higher-grade arteriolar wall thickening. Age, BMI > 25, history of hypertension or diabetes mellitus or dyslipidemia, eGFR (high to low), and SMK/HU (non-SMK/non-HU was used as a reference) were significantly associated with a higher grade of arteriolar hyalinosis in the non-adjusted model. The relationship between SMK/HU and arteriolar wall thickening remained significant even in multivariate model 3, which included age, sex, and potential risk factors such as hypertension and RAS inhibitors (Table 3).

## 4. Discussion

The findings of this study demonstrate the associations of a combination of SMK and HU with renal arteriolar wall thickening but not hyalinosis in IgA nephropathy. In this study, SMK was associated with arteriolar wall thickening only for the existence of HU. Although the confounding effects of advanced age and a history of hypertension or diabetes might be relevant, the relationship between SMK/HU and arteriolar wall thickening was substantial even after adjusting for these confounding factors. Although non-SMK/HU was significantly associated with renal arteriolar hyalinosis, we could not confirm a singular association between SMK and renal arteriolar hyalinosis and wall thickening. With regard to renal outcomes, current SMK was reportedly associated with CKD progression [25,26] and end-stage renal disease [27]. Moreover, a previous study reported on the dose-dependent effect of SMK on renal outcomes, and SMK with more than 20 cigarettes per day was significantly associated with an increased CKD risk [28,29]. A previous study suggested that the risk of arteriolar hyalinosis in patients with IgA nephropathy increased with an increased SMK dose [17,19]. Therefore, the present study may not have enough power to elucidate the sole effects of SMK on renal arteriolar hyalinosis because of insufficient data on cigarette consumption and SMK duration. Meanwhile, an association of HU and arteriolar wall thickening might be common when patients have a history of SMK since a previous study did not show the sole association of HU with wall thickening.

Several mechanisms could underlie the significant association of the combination of HU and SMK with renal arteriolar wall thickening. By activating RAS and the sympathetic nervous system, SMK is linked to hypertension [21]. HU can also be associated with hypertension via the activation of RAS and vascular smooth muscle proliferation [21]. Combining SMK and HU might have a significant influence on arteriolar wall thickening associated with hypertension and RAS activation because renal RAS has been demonstrated to be associated with renal arteriolar wall thickening [30].

Regarding the clinical relevance of the association of the combination of SMK and HU with renal arteriolar wall thickening, the coexistence of HU and SMK could have a great effect on CKD progression. Arteriolar wall thickening, which is a typical finding of nephrosclerosis that can cause renal ischemia [31]. Vasoconstriction by the nicotine-induced activation of the sympathetic nerve [20] can also cause renal ischemia. Therefore, a combination of SMK and HU could accelerate a decline in renal function by inducing renal ischemia, which is the final common pathway to the progression of CKD [32]. In accordance with this notion, Nagasawa et al. [33] reported that the association between SMK and CKD progression was more prevalent at an advanced CKD stage, in which the prevalence of HU was high. The findings of the present study might help in understanding the underlying mechanisms that are responsible for the heterogeneous association of SMK with CKD progression in previous reports. Furthermore, structural changes, including not only functional changes in renal arteriole, could be related to the insufficient recovery of smoking-induced decrease in renal plasma flow by smoking cessation [34].

This study has several limitations. First, we could not confirm whether the combination of SMK and HU induced renal arteriolar hyalinosis because of the cross-sectional design. Second, the results of this study cannot be applied to CKD of any etiology. Renal arteriolosclerosis is commonly observed in patients with diabetes mellitus and/or hypertension. Therefore, the additional effects of SMK and HU on renal arteriolar hyalinosis in patients with CKD remain to be determined. Third, the limited number of renal biopsy samples raises a concern regarding sampling bias. However, we certified a linear correlation between the hyalinosis index and age or blood pressure in this study in accordance with the results of a previous autopsy-based study [35]. Therefore, the effects of sampling bias might have minimal effects on the results. Fourth, the role of uric acid in oxidative stress regulation may have two different aspects. Although uric acid acts as a pro-oxidant inside the cells, it shows antioxidant properties in the extracellular environment [36]. A recent clinical trial using febuxostat, one of the urate-lowering drugs, showed that its treatments for lowering serum UA levels to 5–6 mg/dL had the lowest incidence of renal events [37]. Therefore, the balance between these two contrasting properties could determine its effect on arteriolosclerosis. We could not determine the role of such balances in the clinical setting. Evidently, further studies are warranted to elucidate the prognostic value of SMK and HU and the beneficial effect of SMK cessation on CKD progression when complicated by HU. Fifth, the results might be affected by the exclusion of patients using urate-lowering drugs (n = 6) since their median hyalinosis index was apparently higher than that of the non-users. In fact, the OR for higher arteriolar hyalinosis index of SMK/HU was more than twice that of the non-SMK/HU subgroup when the user of urate-lowering drugs was included in the analysis as HU (data was not shown). Finally, the definition of HU was derived from patients who were Japanese, including those with IgA nephropathy in our previous study [19]. The cause of HU and the cut-off value of UA associated with arteriolar hyalinosis and wall thickening might be different for those of other races and with different kidney diseases. Therefore, the results of this study cannot be generalized.

## 5. Conclusions

We found that SMK affected renal arteriolar wall thickening in the presence of HU but not in the absence of HU in patients with CKD. These associations were independent of age, the history of hypertension or diabetes mellitus, and antihypertensive medication. These findings suggest that the combination of SMK and HU could promote CKD progression through the potentiation of renal arteriolosclerosis. Therefore, large prospective trials are needed to determine the efficacy of interventions such as SMK cessation combined with urate-lowering drugs to retard CKD progression when complicated with HU.

## Figures and Tables

**Figure 1 biomedicines-11-02053-f001:**
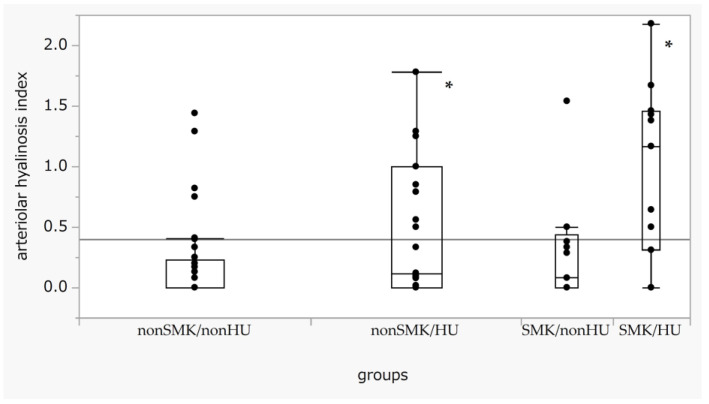
Box-and-whisker plots of the arteriolar hyalinosis index across the SMK/HU groups. Boxes indicate medians with IQRs. Error bar maximum and minimum values defined as the third quartile + 1.5 times the IQR and first quartile − 1.5 times the IQR, respectively. Outliners are defined as date points > 1.5 times the IQR beyond the first or third quartiles. Differences in the arteriolar hyalinosis index across each group were evaluated using the Steel test, with non-SM/non-HU used as a reference. *p* < 0.05 (one-way ANOVA). * *p* < 0.05 compared with non-SMK/non-HU. HU, hyperuricemia; IQR, interquartile range; SMK, smoking.

**Figure 2 biomedicines-11-02053-f002:**
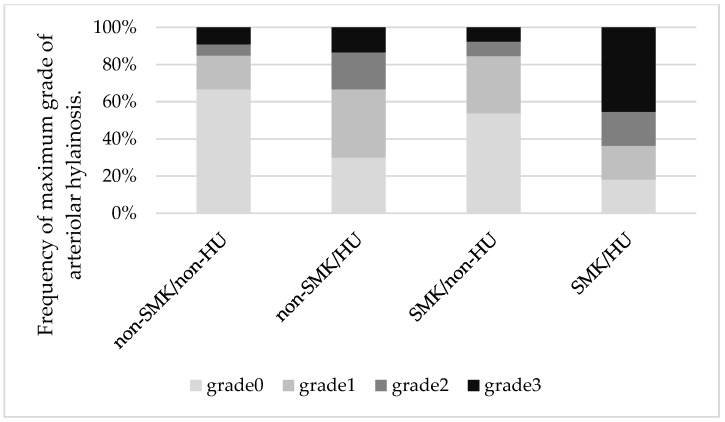
Mosaic plots showing maximum grades of arteriolar hyalinosis across the groups. The rate of a higher max grade for the hyalinosis index (grade 2 + grade 3) was the highest in the SMK/HU subgroup.

**Figure 3 biomedicines-11-02053-f003:**
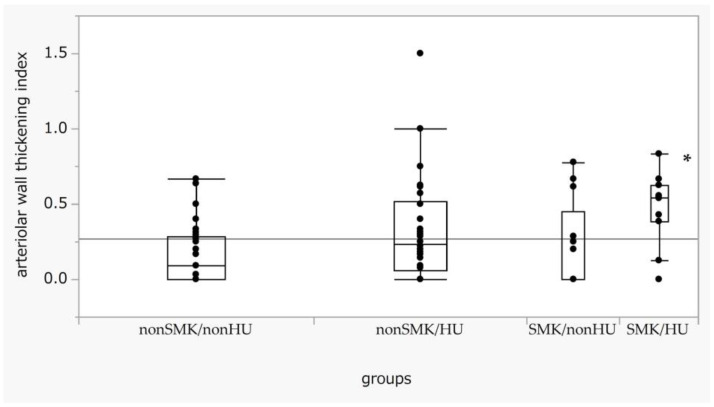
Box-and-whisker plots of arteriolar wall thickening across the SMK/HU groups. Differences in the arteriolar wall thickening index across each group were examined using the Steel test, with non-SM/non-HU used as a reference. *p* < 0.05 (one-way ANOVA). * *p* < 0.05 compared with non-SMK/non-HU. SMK, smoking; HU, hyperuricemia.

**Figure 4 biomedicines-11-02053-f004:**
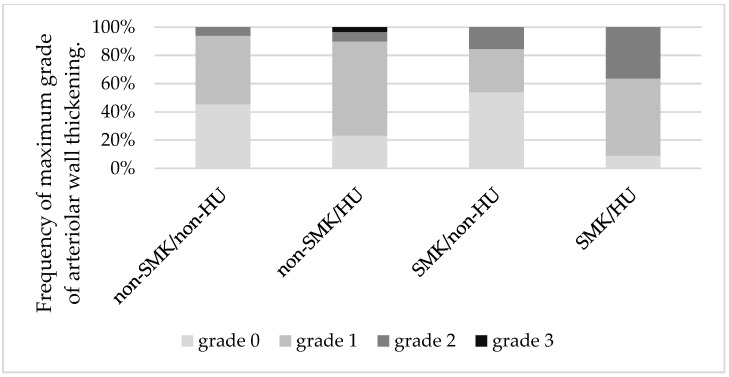
Mosaic plots show the maximum grades of arteriolar wall thickening across the groups. The rate of higher max grade for the arteriolar wall thickening (grade 2 + grade 3) was highest in the SMK/HU subgroup. SMK, smoking; HU, hyperuricemia.

**Table 1 biomedicines-11-02053-t001:** Clinical characteristics of patients in the smoking/hyperuricemia subgroups.

	Non-SMK		SMK		*p*-Value
	non-HU *n* = 33	HU *n* = 30	non-HU *n* = 13	HU *n* = 11	
Men Age (years) BMI (kg/m^2^) Diabetes mellitus Hypertension Dyslipidemia Systolic blood pressure (mmHg) Diastolic blood pressure (mmHg) Proteinuria (g/g creatinine) Serum creatinine (mg/dL) Estimated GFR (mL/min/1.73 m^2^) Serum uric acid (mg/dL) LDL cholesterol (mg/dL) HDL cholesterol (mg/dL) Renal arteriolar hyalinosis, number (%) Renal arteriolar hyalinosis index Diuretics Renin–angiotensin system inhibitors Statins	14 (42) 30 (20–38) 21 (19–23) 1 (3) 5 (15) 7 (21) 120 (110–127) 72 (66–79) 0.63 (0.24–1.1) 0.66 (0.60–0.83) 93.8 (83.2−116) 4.7 (4.2−5.7) 102 (84−127) 60 (49−71) 11 (33) 0 (0−0.2) 0 (0) 7 (21) 0 (0)	6 (20) 32 (22−51) 22 (21−26) 3 (10) 9 (30) 12 (40) 112 (106−129) 70 (68−77) 0.82 (0.39−1.6) 0.68 (0.53−0.93) 85.1 (53.6−124) 6.3 (5.6−7.0) 109 (90−125) 56 (48−71) 21 (70) 0.1 (0−1) 2 (7) 6 (20) 1 (3)	13 (100) 24 (19−52) 23 (19−26) 1 (8) 4 (31) 7 (54) 120 (113−130) 74 (71−82) 0.6 (0.30−1.2) 0.74 (0.71−0.91) 102 (72−119) 6.3 (5.7−6.8) 102 (71−126) 54 (40−67) 6 (46) 0.1 (0−0.4) 0 (0) 4 (31) 2 (15)	8 (72) 49 (38−59) 26 (23−30) 2 (18) 6 (55) 7 (64) 130 (114−138) 80 (70−80) 0.75 (0.49−1.6) 0.88 (0.81−1.2) 69.3 (48.5−90.9) 7.5 (7.1−8.0) 112 (87−130) 49 (40−64) 9 (82) 1.2 (0.3−1.5) 0 (0) 3 (27) 1 (9)	<0.0001 0.02 0.003 0.1884 0.0203 0.0342 0.1713 0.4006 0.2550 0.0129 0.04006 <0.0001 0.723 0.428 0.0058 0.0002 0.2737 0.8569 0.1297

Data are expressed as medians (IQR) or numbers (%). BMI, body mass index; eGFR, estimated glomerular filtration rate; HDL, high-density lipoprotein; HU, hyperuricemia; IQR, interquartile range; LDL, low-density lipoprotein; SMK, smoking. Hyperuricemia was defined as uric acid levels of ≥7 mg/dL in men and ≥5 mg/dL in women.

**Table 2 biomedicines-11-02053-t002:** Multivariate-adjusted odds ratios for higher arteriolar hyalinosis index.

	Model 1	Model 2	Model 3
	OR (95% CI)	OR (95% CI)	OR (95% CI)
non-SMK/non-HU	1.00 (reference)	1.00 (reference)	1.00 (reference)
non-SMK/ HU	5.60 * (1.26–24.9)	6.10 * (1.07–34.7)	7.62 * (1.17–49.5)
SMK/non-HU	0.51 (0.07–4.07)	0.43 (0.04–4.40)	0.57 (0.06–5.70)
SMK/HU	4.85 (0.72–32.7)	4.53 (0.57–37.0)	6.13 (0.64–58.3)

Abbreviations: CI, confidence interval. OR, odds ratio; SMK, smoking; HU, hyperuricemia. A higher arteriolar hyalinosis index was defined as ≥0.41. Variables were used for adjustment. Model 1: Age and sex. Model 2: Model 1 + comorbidities (diabetes mellitus, dyslipidemia, hypertension, body mass index > 25) and eGFR (mL/min/1.73 m^2^). Model 3: Model 2 + renin–angiotensin system inhibitors * *p* < 0.05.

**Table 3 biomedicines-11-02053-t003:** Multivariate adjusted odd ratios for higher arteriolar wall thickening index.

	Model 1	Model 2	Model 3
	OR (95% CI)	OR (95% CI)	OR (95% CI)
non-SMK/non-HU	1.00 (reference)	1.00 (reference)	1.00 (reference)
non-SMK/HU	1.77(0.49–6.40)	0.99 (0.23–4.28)	1.12 (0.25–5.08)
SMK/non-HU	1.42 (0.20–10.2)	1.5 (0.16–13.9)	2.11 (0.21–20.9)
SMK/HU	13.0 * (1.77–95.1)	10.2 * (1.19–88.0)	12.8 * (1.36–119)

Abbreviations: CI, confidence interval. OR, odd ratio; SMK, smoking; HU, hyperuricemia. A higher arteriolar wall thickening index was defined as equal or greater than 0.316. Variables were used for adjustment. Model 1: Age and sex. Model 2: Model 1 + comorbidities (diabetes mellitus, dyslipidemia, hypertension, body mass index > 25) and eGFR (mL/min/1.73 m^2^). Model 3: Model 2 + renin–angiotensin system inhibitors. * *p* < 0.05.

## Data Availability

The data presented in this study are available on request from the corresponding author. The data are not publicly available due to the Ethics Committee permission.

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
