# Peer review of "The Association of Smoking and Hyperuricemia with Renal Arteriolosclerosis in IgA Nephropathy"

_biomedicines, 2023, doi:10.3390/biomedicines11072053_

Round 1

Reviewer 1 Report

Scientific merit of this well presented observational study is minor.

Study limitations are mentioned on p. 8. However, the controversial role of HU in terms of oxidative stress enhancement/suppression must be discussed, not just mentioned. The confusing formulation of SMK-/HU+ vs. SMK-/HU+ must be corrected.

The reference list should be abbreviated, especially the papers over 20 years old. The number of authors quoted must be limited to 3 or 6, consistently, and the doi codes added to all or to none of the quotations.

Author Response

Responses to Reviewer Comments:

We appreciated for reviewing our manuscript and giving many valuable comments. Responses to each comment are following. Words highlighted in red indicate modifications (changes or additions) during revision.

Responses to the comments from Reviewer 1

Comments 1

Study limitations are mentioned on p. 8. However, the controversial role of HU in terms of oxidative stress enhancement/suppression must be discussed, not just mentioned.

Reply

Thank you for your very valuable comment. We have added following discussion of this issue with references.

Revised, P8, L30

Forth, the role of uric acid for oxidative stress may have two different aspects. Although uric acid acts as a prooxidant inside the cells, it shows rather antioxidant property in the extracellular environment.[41] Recent clinical trial using febuxostat, one of the urate-lowering drugs showed that its treatments for serum uric acid lowering levels of 5-6 mg/dL had the lowest incidence of renal events.[42] Therefore, the balance between these two different properties may determine its effect on arteriolosclerosis. We cannot determine the role of such balances in the clinical settings. Obviously, further studies are warranted to elucidate the prognostic value of SMK and HU and the beneficial effect of SMK cessation on CKD progression complicated by HU.

 Comments 2

The confusing formulation of SMK-/HU+ vs. SMK-/HU+ must be corrected.

Reply

Thank you for the suggestion. We have corrected the formulation such SMK/HU and non-SMK/non-HU in the whole of the manuscript.

Comments 3

The reference list should be abbreviated, especially the papers over 20 years old. The number of authors quoted must be limited to 3 or 6, consistently, and the doi codes added to all or to none of the quotations.

Reply

We changed the references over 20 years old to newer ones as possible as we can. And we collected the reference list according to the suggestion.

Reviewer 2 Report

Comments of this reviewer are:

Main Comment

“Therefore, we analyzed the prevalence of HU, defined as UA levels of >5 mg/dL in women and >7 mg/dL in men or the use of urate-lowering drugs.” The use of urate-lowering drugs may (or may not) affect the arterial hyalinosis and may bias the effect of true HU (> 5 or 7 mg/dL). In addition, if we consider that HU affect arteriosclerosis in IgAN in this study, in case that you include in the HU group those on urate therapy, the conclusion is that the urate-lowering therapy is worthless as even those in therapy have more arteriosclerosis.  Thus, I would suggest redoing the analysis excluding patients on urate-lowering therapy from the study. (Including only those in urate-lowering therapy that consistently had urate levels > 5 or 7 mg/dL is not an alternative, as I think it is very difficult to identify these patients in a cross-sectional study)

Other comments

- “[18]{Zamami, 2016, Modification of the impact of hypertension on proteinuria by renal arteriolar hyalinosis in nonnephrotic chronic kidney disease}” mention only the refs’ no

- In the introduction you provide the rationale for the association of smoking with arteriosclerosis.  However, you need to provide in the introduction the relative rationale for any association of hyperuricemia with arteriosclerosis to justify why you have chosen to study hyperuricemia in your paper.

- The limited generalizability is another limitation as it refers only in Japanese patients with definition of HU based in this specific population

Author Response

Responses to Reviewer Comments:
We appreciated for reviewing our manuscript and giving many valuable comments. Responses to each comment are following. Words highlighted in red indicate modifications (changes or additions) during revision. 

More details please see the attachment. 

Round 2

Reviewer 2 Report

Comments of this reviewer are:

I would suggest replacing the analysis with the new one, excluding patients on urate therapy.

Author Response

Responses to Reviewer Comments:

We appreciated for reviewing our manuscript and giving many valuable comments. Responses to each comment are following.

Responses to the comments from Reviewer

Comment

  1. I would suggest replacing the analysis with the new one, excluding patients on urate therapy.

Reply

>> According to your suggestion, we have analyzed similar analysis excluding user of urate lowering therapy (n=6). Based on the results of reanalysis, we revised the manuscript as following.

Abstract

P1, L11

We examined the cross-sectional association between smoking (current or former) and renal arteriolar hyalinosis and wall thickening with or without HU [uric acid (UA) level ≥ 7 and ≥ 5 mg/dL in men and women] in 87 patients with IgA nephropathy who underwent renal biopsy.

P1, L17

Multiple logistic analysis showed that SMK/HU, but not SMK/non-HU was significantly associated with an increased risk of higher-grade renal arteriolar wall thickening, but not hyalinosis than with non-SMK/non-HU. The adjusted odds ratio (95% confidence interval, P value) for SMK/HU was 12.8 (1.36-119, P < 0.05) for wall thickening. An association between SMK and renal arteriolar wall thickening may be prevalent only among patients with HU in patients with IgA nephropathy.

Material and Methods

P2, L19

The six patients using urate-lowering drugs were excluded in the analysis. Finally, 87 patients were included in this study.

P2, L47

Therefore, we defined as UA levels of ³5 mg/dL in women and ³7 mg/dL in men.

P3, L10

Higher-grade arteriolar hyalinosis and wall thickening were defined as hyalinosis or wall thickening with values equal to or greater than the third tertile hyalinosis index (0.41) and wall thickening index (0.316).

Result

P3

Table 1. Clinical characteristics of patients in the smoking/hyperuricemia subgroups

               non-SMK                     

          SMK     

P-
value

non-HU                  HU

n = 33              n = 30

non-HU                 HU

n = 13             n = 11 

Men

Age (years)

BMI (kg/m2)

Diabetes mellitus

Hypertension

Dyslipidemia

Systolic blood pressure (mmHg)

Diastolic blood pressure (mmHg)

Proteinuria (g/g creatinine)

Serum creatinine (mg/dL)

Estimated GFR (mL/min/1.73m2)

Serum uric acid (mg/dL)

LDL cholesterol (mg/dL)

HDL cholesterol (mg/dL)

Renal arteriolar hyalinosis, number (%)

Renal arteriolar hyalinosis index

Diuretics

Renin–angiotensin system inhibitors

Statins

14 (42)

30 (20-38)

21 (19-23)

1 (3)

5 (15)

7 (21)

120 (110-127)

72 (66-79)

0.63 (0.24-1.1)

0.66 (0.60-0.83)

93.8 (83.2-116)

4.7 (4.2-5.7)

102 (84-127)

60 (49-71)

11 (33)

0 (0-0.2)

0 (0)

7 (21)

0 (0)

6 (20)

32 (22-51)

22 (21-26)

3 (10)

9 (30)

12 (40)

112 (106-129)

70 (68-77)

0.82 (0.39-1.6)

0.68 (0.53-0.93)

85.1 (53.6-124)

6.3 (5.6-7.0)

109 (90-125)

56 (48-71)

21 (70)

0.1 (0-1)

2 (7)

6 (20)

1 (3)

13 (100)

24 (19-52)

23 (19-26)

1 (8)

4 (31)

7 (54)

120 (113-130)

74 (71-82)

0.6 (0.30-1.2)

0.74 (0.71-0.91)

102 (72-119)

6.3 (5.7-6.8)

102 (71-126)

54 (40-67)

6 (46)

0.1 (0-0.4)

0 (0)

4 (31)

2 (15)

8 (72)

49 (38-59)

26 (23-30)

2 (18)

6 (55)

7 (64)

130 (114-138)

80 (70-80)

0.75 (0.49-1.6)

0.88 (0.81-1.2)

69.3 (48.5-90.9)

7.5 (7.1-8.0)

112 (87-130)

49 (40-64)

9 (82)

1.2 (0.3-1.5)

0 (0)

3 (27)

1 (9)

<0.0001

0..02

0.003

0.1884

0.0203

0.0342

0.1713

0.4006

0.2550

0.0129

0.04006

<0.0001

0.723

0.428

0.0058

0.0002

0.2737

0.8569

0.1297

P5

Figure 1. Box-and-whisker plots of the arteriolar hyalinosis index across the SMK/HU groups. Boxes indicate medians with IQRs. Error bar maximum and minimum values defined as the third quartile + 1.5 times the IQR and first quartile − 1.5 times the IQR, respectively. Outliners are defined as date points > 1.5 times the IQR beyond the first or third quartiles. Differences in the arteriolar hyalinosis index across each group were evaluated using the Steel test, with non-SM/non-HU used as a reference. P < 0.05 (one-way ANOVA). *P < 0.05 compared with non-SMK/non-HU. HU, hyperuricemia; IQR, interquartile range; SMK, smoking.

Figure 2. Mosaic plots showing maximum grades of arteriolar hyalinosis across the groups. The rate of higher max grade of hyalinosis index (grade 2 +grade 3) was the highest in the SMK/HU subgroup.

P6

Figure 3. Box-and-whisker plots of arteriolar wall thickening across the SMK/HU groups. Differences in the arteriolar wall thickening index across each group were examined using the Steel test, with non-SM/non-HU used as a reference. P <0.05 (one-way ANOVA). *P<0.05 compared with non-SMK/non-HU. SMK, smoking; HU, hyperuricemia.

Figure 4. Mosaic plots show maximum grades of arteriolar wall thickening across the groups. The rate of higher max grade of arteriolar wall thickening (grade 2+grade3) was highest in the SMK/HU subgroup. SMK, smoking; HU, hyperuricemia.

P6

3.3. The risk of Higher-grade Arteriolar Hyalinosis and Wall Thickness

We analyzed the odds ratios (ORs) of various risk factors for higher-grade arteriolar hyalinosis. Age, history of hypertension or diabetes mellitus or dyslipidemia, eGFR (high to low), SMK/HU and non-SMK/HU, (non-SMK/non-HU was used as a reference) were significantly associated with a higher grade of arteriolar hyalinosis in the non-adjusted model. The association between non-SMK/HU and arteriolar hyalinosis was significant in the multivariate model 3, which included age, sex, potential risk factors such as hypertension and the use of RAS inhibitors (Table 2).

P7

Table 2. Multivariate-adjusted odds ratios for higher arteriolar hyalinosis index (excluding the patients using urate lowring drugs)

Model 1

Model 2

Model 3

OR (95% CI)

OR (95% CI)

OR (95% CI)

non-SMK/non-HU

1.00 (reference)

1.00 (reference)

1.00 (reference)

non-SMK/HU

5.60* (1.26–24.9)

5.25 (0.92-30.0)

6.82* (1.04-44.8)

SMK/non-HU

0.51 (0.07–4.07)

0.45 (0.05-4.40)

0.59 (0.06-5.73)

SMK/HU

4.85 (0.72–32.7)

3.94 (0.48-32.2)

5.60 (0.57-54.7)

Table 3. Multivariate adjusted odd ratios for higher arteriolar wall thickening index.

Model 1

Model 2

Model 3

OR (95% CI)

OR (95% CI)

OR (95% CI)

non-SMK/non-HU

1.00 (reference)

1.00 (reference)

1.00 (reference)

non-SMK/HU

1.77(0.49-6.40)

0.99 (0.23-4.28)0

1.12 (0.25-5.08)

SMK/non-HU

1.42 (0.20-10.2)

1.5 (0.16-13.9)

2.11 (0.21-20.9)

SMK/HU

13.0* (1.77-95.1)

10.2* (1.19-88.0)

12.8* (1.36-119)

P8, L1

  1. Discussion

The findings of this study demonstrated associations of a combination of SMK and HU with renal arteriolar wall thickening, but not hyalinosis in IgA nephropathy. In this study, SMK was associated with arteriolar wall thickening only in the existence of HU. Although the confounding effects of advanced age and a history of hypertension or diabetes may be relevant, the relationship between SMK/HU and arteriolar wall thickening was substantial even after adjusting for these confounding factors. Although non-SMK/HU was significantly associated with renal arteriolar hyalinosis, we could not confirm a singular association between SMK and renal arteriolar hyalinosis and wall thickening.

P8, L9

Several mechanisms may underlie the significant association of the combination of HU and SMK with renal arteriolar wall thickening. By activating RAS and the sympathetic nervous system, SMK is linked to hypertension [21]. HU is also associated with hypertension via activation of RAS and vascular smooth muscle proliferation [21]. Combining SMK and HU may have a significant influence on arteriolar wall thickening associated with hypertension and RAS activation because renal RAS has been demonstrated to be associated with renal arteriolar wall thickening [30].

P8, L16

Regarding the clinical relevance of the association of the combination of SMK and HU with renal arteriolar wall thickening, the coexistence of HU and SMK may have much effect on CKD progression.

P8, L20

Therefore, a combination of SMK and HU can accelerate declining renal function by inducing renal ischemia, which is the final common pathways to the progression of CKD [32].

P8, L48

Fifth, the results might be affected by exclusion of patients using urate-lowering drugs (n=6), since their median hyalinosis index was apparently higher than that of the non-users. In fact, the OR for higher arteriolar hyalinosis index of SMK/HU was more than twice that of the non-SMK/HU subgroup when the user of urate-lowering drugs were included in the analysis as HU (data was not shown).

Conclusion

P9, L6

We found that SMK affected renal arteriolar wall thickening in the presence of HU, but not in the absence of HU, in patients with CKD.

Round 3

Reviewer 2 Report

The authors addressed all comments in the response to the reviewers letter.  However, the manuscript versions available online are all with the 93 patients and not with 87 patients.  Just, upload the correct version

Author Response

A new version, based on an analysis of 87 patients, appears to have already been uploaded online. We will upload the new version again.
